# A Review of the Energy Policies of the BRICS Countries: The Possibility of Adopting a Just Energy Transition for South Africa

Rajesh Ramluckun [1,*], Nandi Malumbazo [1] and Lwazi Ngubevana [2]

1   School of Chemical and Metallurgical Engineering, University of the Witwatersrand, 1 Jan Smuts Avenue, Johannesburg 0001, South Africa; nandi.malumbazo@wits.ac.za
2   African Energy Leadership Centre, Wits Business School, 2 St Davids Place and St Andrews, Johannesburg 2193, South Africa; lwazi.ngubevana@wits.ac.za
*   Correspondence: rajesh.ramluckun@icloud.com

**Abstract:** This review focuses on the Just Energy Transition (JET) policies of the BRICS member countries with the aim of finding lessons and possibly adopting some of the key energy policy aspects utilised in other countries as a benchmark for the South African context. We consider the present stage of JET in South Africa while being cognisant of the energy source mix supporting the electricity sector and the lifespan and condition of its power plants. An analysis of the energy mix revealed that all nations are dependent on electricity for energy, which is produced predominantly from fossil-fuelled power plants with high GHG emissions (concentrating on high $CO_2$ emissions). It was concluded that some of the lessons learnt from the BRICS countries might be incorporated into a South African energy plan for the transition, with options and guidance for the formulation of policies. This study will attract a responsible, environmentally conscious audience that has the commitment and drive to combat global warming and curb climate change.

**Keywords:** BRICS; Just Energy Transition (JET); policy; renewable energy; climate change; power generation

## 1. Introduction

At its inception in 2006, there were four BRIC member countries comprising Brazil, Russia, India and China. In 2010, South Africa was invited to become a member of BRIC, which then evolved into BRICS [1], forming a new alliance of these five member countries. South Africa joined in December 2010 to promote peace, security, economic development and cooperation amongst these countries [2]. The main objective was to contribute significantly to improving the socio-economic landscape of these five member countries, which have over three billion people. This is approximately 40% of the global population, and they account for approximately 25% of the world's gross domestic product (GDP). International growth, particularly in developing countries, is positively impacted by the sheer magnitude of the BRICS member countries' geodemographics and summated economy [1]. BRICS also assists in curbing and softening adverse impacts during global financial challenges such as in recessions by advocating resilience in commerce [3].

The objectives, advancements and successes of the BRICS member countries' energy plans can be objectively measured with their Energy Transition Index (ETI) ratings (with a range from 0 to 100, the higher the better in this instance) and Just Energy Transition (JET) rankings (the lower the better in this instance) published by the World Economic Forum (WEF). The BRICS countries' JET rankings and ETI ratings, respectively, are 30 and 66 for Brazil, 73 and 56 for Russia, 87 and 53 for India, 68 and 57 for China, and 110 and 48 for South Africa [4]. These ratings and rankings offer a direct correlation for evaluating the robustness and flexibility of the JET. For a resilient or robust transformation, the movement of the JET must maintain forward transitional momentum in order to achieve

dependable, cost-effective and self-sustaining energy systems in the face of adversity, such as the COVID-19 pandemic that the world experienced.

A review of BRICS member countries' NDCs was conducted to assess the efforts made to reduce climate change and the goals aimed at an ultimate achievement of net-zero $CO_2$ emissions in BRICS countries. The aim was to identify policies and measures executed to influence green employment, such as policies on social protection, organizational expansion, occupational safety and health, communication, skills, education and training in cognisance of the ILO Guidelines for a Just Transition [5].

All BRICS countries have pledged their commitment to net-zero targets with submissions of their NDCs as further confirmation (Brazil by 2060, Russia by 2060, India by 2070, China by 2060 and South Africa by 2050). Brazil and Russia submitted their NDCs in the year 2020, China's and South Africa's NDCs were submitted in the year 2021 and the NDCs from India were submitted in 2016 [6]. A common observation of the BRICS countries' NDCs is that social elements are vaguely referenced. The role of or need for social partners or social dialogue is not covered either. Apart from South Africa, none of the other BRICS countries addressed Just Transition as part of the Just Energy Transition (JET) and heat stress aspects. However, the Russian NDCs mention the term "fairness" and are cognisant of the social impacts of climate change and JET. India refers to jobs and power eradication whilst bearing in mind diverse groups' specific weaknesses to climate change. China's NDCs emphasise few employment consequences of climate change, with the requirement to foster an equitable, reasonable, cooperative and win–win global climate policy [7]. All BRICS countries' NDCs have considered, to an extent, education, training and skilling, but more attention to detail must be paid to the reskilling and upskilling of nationwide workforces in all other countries' NDCs apart from South Africa. It was also revealed in a detailed assessment of the financial and environmental aspects that financial development was integrated together with environmental emissions and resource use. The study recommended separating economic expansion and development from $CO_2$ emissions and resource usage [7].

The aim and objective of this review is to assess BRICS's energy policies or those of the individual member countries and, thereafter, make recommendations for South Africa, considering lessons and potentially adopting relevant energy policy aspects implemented in other countries as a benchmark for the South African context. We consider the present stage of JET in South Africa and the energy source mix supporting the electricity sector, being cognisant of the lifespan and condition of existing power plants.

This paper is divided into several sections. The next four sections provide an overview of the energy policies of BRICS member countries such as India (Section 2), China (Section 3), the Russian Federation (Section 4) and Brazil (Section 5). Section 6 represents the findings of a review of the energy policy in South Africa in the context of ensuring a Just Energy Transition. Finally, in Section 7, conclusions and prospects for further research are described.

## 2. India

### 2.1. Individual Indicators of India's Macroeconomics

India's economic growth in 2022 was 6.7%, with an anticipated GDP of 5.6% for the years 2023 to 2024. Currently, India is ranked 59 out of 152 countries in the GlobalData Country Risk Index [8]. A direct result of the present regulations of the Securities and Exchange Board of India (SEBI) is that India's equity market has become world-class and attractive due to concerted efforts towards regulation and liberalization. According to the World Bank, local organisations, listed as a constituent of GDP, increased from 80.8% in 2019 to 97.3% in 2020 in India, thereby increasing the country's equity significantly and exceeding China's percentage of GDP, being 83.2% [9].

### 2.2. India's Energy Policy

India's current energy mix comprises 46% coal, 25% petroleum (and other liquid fuels), 20% biomass and waste, 6% natural gas, 1% nuclear, 1% hydroelectric and 1% other

sources of renewable energy, as shown in Figure 1 [10]. The social impact on the Indian population is a key focus and the energy plan enforces it as a pivotal point to transition harmoniously. It considers the quality of life of low-income rural communities, who make up approximately 700 million people and are forced to use biomass as fuel to cook their meals [10]. The Prime Minister of the Indian Government, Narendra Modi, announced that India is committed to zero $CO_2$ emissions by 2070 at the COP 26 meeting held in Glasgow in 2021 [11].

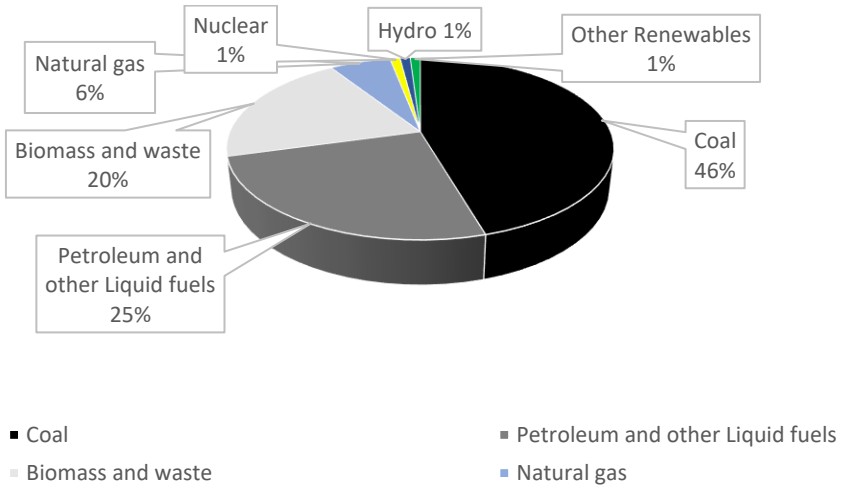

**Figure 1.** India's current energy mix.

A draft version of the country's new National Electricity Plan (NEP) from the Central Electricity Authority (CEA) was published for communal participation and includes an appraisal of the detailed electricity capacity expansion plans for 2032 [12]. The predicted expansion of generating power plants will be approximately 40%, from 623 GW to 866 GW. The bulk of the increase of 214 GW is expected to be from renewable energy sources, with approximately 69% contributed by solar photovoltaics (PV). An additional 10 GW from an offshore wind power plant is also expected by 2032. Therefore, renewable energy is expected to contribute a total of 60% (approximately 516 GW) of the total electricity supply by 2032. The contribution of fossil fuel electricity production is expected to decline by 2032, from 46% to 28%. Gas power plants' installed capability is projected to decrease to 3% by 2032. The hydro-electric power contribution of 69 GW is also expected to increase to 7% by 2032 as depicted in Figure 2. A nuclear power plant capacity of 22.5 GW will contribute 2.5% to the electricity pool [13].

The unavailability of affordable electricity results in poor air quality (rural populations resort to using easily obtainable fuel sources like wood for open fires to prepare meals and for heating during colder seasons), which negatively impacts the health of low-income rural communities and the environment. This was exacerbated by the COVID-19 pandemic and financially constrained the electricity sector. In the year 2021, the air quality in some cities in India rated among other similarly positioned countries as the poorest globally due to the distribution of utility companies using coal, petroleum and other liquid fuels (having low efficiency and high $CO_2$ emissions) as major electricity generation sources [13]. The country, with its population of one billion, is the world's third-largest electricity consumer and is fed by 71% fossil fuel (coal 46%, petroleum and other fossil fuels 25%) power plants [10]; however, in compliance with energy policy, India has approved budgets and committed to transition towards 50% renewable energy by 2030 [14].

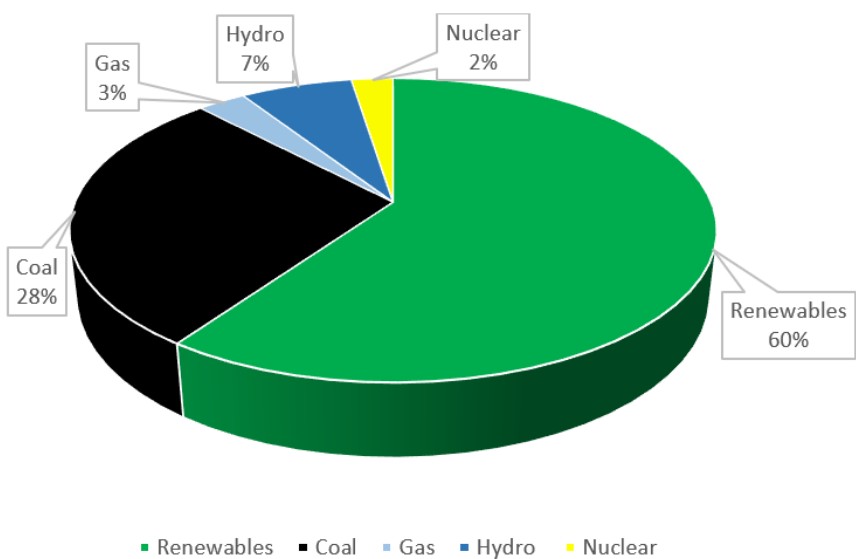

**Figure 2.** India's envisioned 2032 energy mix.

India has adopted an implementation strategy where 38% of the total energy market will be from renewable energy sources with an emphasis on solar energy technologies [15]. The urgency spearheading the international reaction to climate-related change is a fundamental priority which India has committed to. India is seen as a huge participant in the world's energy ecosystem, with energy consumption that has doubled the energy demand from year 2000. Commendable success was achieved in 2019 when electricity connections were completed to meet the demand of approximately 900 million citizens [10]. The global COVID-19 pandemic has delayed progress towards environmentally friendly fuel technology aimed at reducing greenhouse gases (GHG); however, the pandemic resulted in a slow economic growth of 35% in comparison with a forecasted economic growth of 50% between the years 2019 and 2030, which will contribute to delaying the impact of and transition to green fuels [16].

The sustainable development case shows how India could increase its green energy investment to produce a faster peak, resulting in a decline in GHG emissions in synergy with zero-$CO_2$ emissions targets [16]. This would be a positive impact for speedily attaining the Sustainable Development Goals (SDG). It is important to reiterate that the economic growth of the country has been led predominantly by the nonmanufacturing sector instead of the electricity-dependant industrial and commercial sector. This has slowed down the rate of industrialisation and urbanisation, but it is about to change, as 270 million people are likely to become urbanised in the next 20 years, giving rise to exponential growth in the infrastructure required to support this parabolic increase [16]. The expansion would also support the ILO's 2015 policy as huge infrastructure projects will increase the country's labour requirements, thereby increasing the employment rate.

Over the past two decades, the power generation mix for India's sustainable development plan for cost-efficient and more affordable renewable energy generation of 450 GW by 2030 has been on target [10]. To meet the planned targets for moving forward with new power plants (fossil fuel or renewables), design considerations of robustness, reliability, dependability, flexibility and cost effectiveness will be factored in [17]. Currently, India's electricity grid and its fossil fuel power plants meet most of India's needs, and, supplemented by hydroelectric and gas power plants, grid stability is achieved particularly during power-swing conditions. This ensures the quality of the power supply, with voltage, frequency and reactive power control to sustain and protect the power system from failure. Battery energy storage systems (BESS) are ideal to render the interim flexibility or resilience that India requires to meet the country's solar-driven midday maximum electricity production and early evening maximum load requirements, from which a planned capacity of 140 GW is supplied [10].

India's NDC was tendered to the UNFCCC in the year 2016. India did not present an update to the year 2030 target; however, it is expected to exceed the target with its current policies [18]. During COP26 in Glasgow in 2021, India presented a net-zero target by 2070. But again, like for the other BRICS countries, the NDC does not categorically refer to just transition, decent employment, conditions of work, OHS, social protection or social dialogue. However, the NDC mentions climate and social fairness specifically with regards to improving the rural population's livelihoods, with an emphasis on disadvantaged sectors of the population [5]. Likewise, the NDC contains some organisations' initiatives that concentrate on resource efficiency, waste management, contamination handling, and the public employment initiative of the Mahatma Gandhi National Rural Employment Guarantee Act (MGNREGA). The MGNREGA is an excellent model of how community work initiatives integrate with environmental aims: the MGNREGA's objective is to aid social protection and economic defence to eradicate poverty in rural communities, supporting drought proofing, flood management and the enabling of disadvantaged societies. The MGNREGA states that every countryside home is entitled to 100 days of work per annum. The Ministry of Rural Development has indicated that 60% of the labour supplied by MGNREGA in the year 2012 was predominantly in water conservation and 12% was related to the provision of irrigation services [19]. Female labourers employed in household domestic jobs are rewarded with greater compensation than those engaged in other countryside occupations [20]. India's NDC does mention the need for funding in education and the enhancement of expertise across disciplines as well as for "establishing more intensive state-centric knowledge" on various attributes of renewable energy. Funding in research and development institutes for pre-competitive research is also integrated into India's NDC, which is vital to reach the zero-$CO_2$ mark in compliance with the Paris Climate Change Agreement [20].

In conclusion, India's commitment to the JET plans is further reinforced by the G7 countries, including Norway, Denmark and the European Union (EU). The Just Energy Transition Partnership (JETP) will financially empower India to reduce climate-changing emissions from power production. India reiterated the country's commitment to contributing to eradicating global warming by achieving the following Just Energy Transition objectives/goals [10].

- A target of 500 GW non-fossil electricity capacity by 2030;
- A renewable energy contribution of 50% of electricity demands by 2030;
- A forecasted decrease in $CO_2$ emissions by one billion tonnes from present to 2030;
- A decline by 45% of the $CO_2$ intensity of the economy by 2030, referenced with 2005 emissions;
- A zero-carbonisation goal by 2070.

## 3. China

### 3.1. Individual Indicators of China's Macroeconomics

The real GDP growth of China was 4.5% in 2022, and it is predicted to be 4.8% in 2023, while consumption (use of goods and services) was 6.5% in 2022 (Macroeconomic Outlook Report: China, 2023). The World Bank conducting business rank for China in 2022 was 31 out of 190 countries. China's electrical and electronic trade magnitude is forecasted to grow at a compound annual growth rate (CAGR) of 11.6% from year 2021 to 2025 to a targeted monetary amount of $3.3 trillion in the year 2025 [21].

### 3.2. China's Energy Policy

The current energy mix of China comprises 52% coal, 3% solar, 2% biomass, 10% natural gas, 3% nuclear, 8% hydro, 4% wind and 18% oil as shown in Figure 3. China's planned transition by 2060 would result in a composition of 3% coal, 23% solar, 5% biomass, 3% natural gas, 19% nuclear, 15% hydro, 24% wind and 8% oil by 2060 as depicted in Figure 4. With the Accelerated Transition Scenario (ATS) and the Announced Pledges Scenario (APS), China has been adding nuclear (70 GW) (20 new reactors) and solar (110 GW) capacity to cater for the requirements of 33 million homes [22].

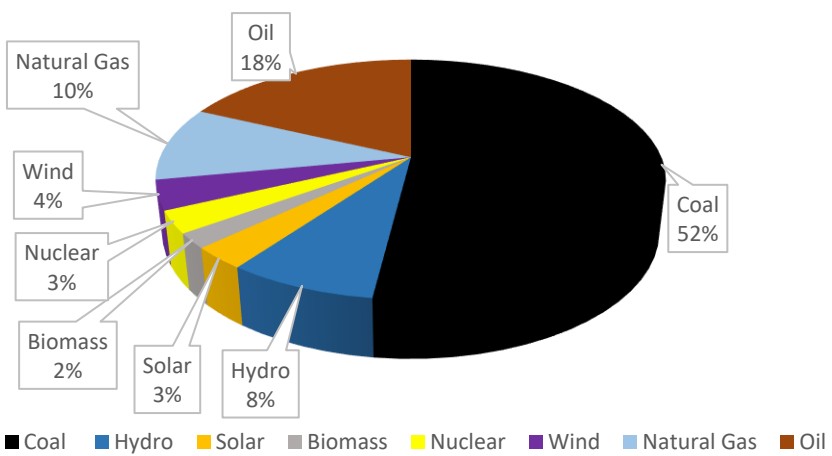

**Figure 3.** China's current energy mix.

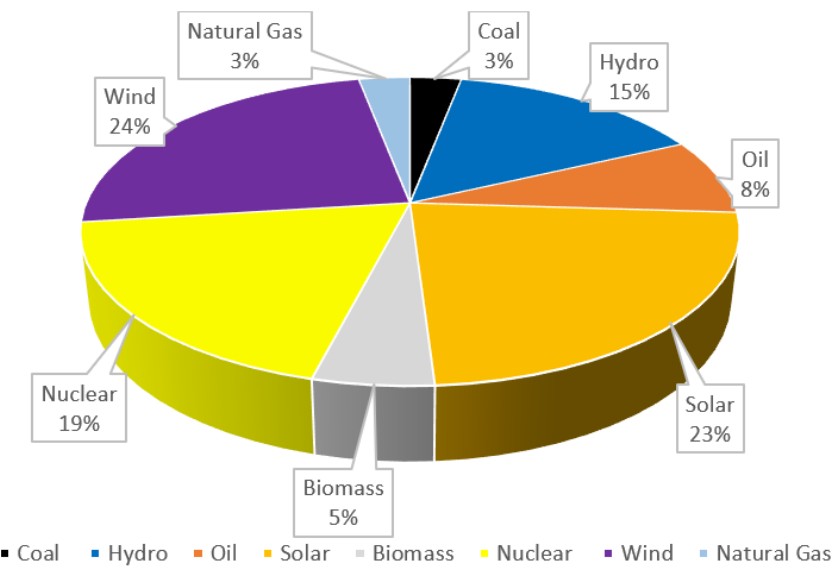

**Figure 4.** China's envisioned energy mix by 2060.

China has been a major player in the world energy market since becoming the world's biggest energy consumer in 2009 [23]. The country is ranked first on the world carbon emissions rankings and is one of the world's leading $CO_2$ producers, contributing 33% of the world's $CO_2$ discharge [22]. Power-producing plants are the origin of approximately 90% of the country's GHG, highlighting that energy plans are imperative for carbon neutrality [23]. China believes in collaborative engagements with the IEA and other countries with the common vision of eliminating carbon emissions. The spin-off from achieving net-zero carbonisation is envisaged to foster China's sustainable economic and environmental development goals while cultivating prosperity in improving technology leadership, resulting in sustainable economic growth led by innovation [23]. One of China's noteworthy achievements and contributions globally is that it produces and supplies 70% of electric vehicle battery demand. The rapid rate of solar plant deployment has mitigated their environmental impact to being the second-largest consumer of oil globally. It must also be noted that China is the biggest supplier of steel and cement, which makes up more than 50% of the global demand. The impact is that the carbon emissions generated by these plants are astronomical and are deemed to be more than the European Union's carbon emissions [23].

The major contributors to the country's $CO_2$ emissions include the electricity utility sector, where 48% of $CO_2$ production is from energy and industrial activities; 36% is

from the steel and cement industry; 8% is from the transport sector; and 5% is from infrastructure (buildings). China has a five-year plan termed the Modern Energy System, which was formulated to reduce energy intensity by 13.5% (from 65% to 51.5%), leading to the reduction of $CO_2$ intensity by approximately 18% and to integrated energy systems where the focus is to grow the renewable energy market percentage to 20% of the country's entire energy consumption by the year 2025 [24]. It is imperative for China to achieve these short-term policy targets, as it will show the IEA and global authorities that the country's $CO_2$ emissions from coal are on target since 2020, i.e., for achieving 15.9% non-fossil fuels, and that a modest decline will be reached by the year 2030. China's pledge in 2021 at the United Nations General Assembly (UNGA) reiterates the country's determination to end the construction of coal-fired power stations in solidarity with a greener economy in the energy sector. Linked to their economic growth forecast, it is predicted that energy demand will remain stable or in line with the economy due to efficiency gains in the energy transition to cleaner fuel sources, which will in turn lead to an improvement in air quality [23]. China's future energy mix will be dominated by solar and hydrogen, being at least 60% of the total energy requirement by the year 2045. The emphasis will be to reduce the dependency on fossil fuels, especially coal, by 80%, oil by 60% and natural gas by more than 45% by 2060 [23]. China is in a favourable financial position to fund the transformation to clean energy and infrastructure with a committed budget of $640 billion to the year 2030 and $900 billion from the year 2031 to the year 2060 [22]. In comparison, the annual energy investment share of the country's GDP was at an average of 2.5% in the years 2016 to 2020, which will more than halve to approximately 1.1% by the year 2060 [24].

Furthermore, another target is to decrease $CO_2$ emissions in the construction sector by at least 95% of current emissions by the year 2060 through electrification and solar-powered climate control (air conditioning and innovative energy-reduction concepts in building designs) to make buildings less energy-intensive [22]. Synchronising the level of the emissions' maximum and also the rate of emissions reduction at the time that the maximum is attained are paramount for the success of China's ultimate objective of zero $CO_2$ emissions. China is fortunate to have the technical knowledge, economic strength and sound policy to accomplish efficient green energy transition by the year 2030 [25].

The energy policy's success would result in a rapid reduction in the fossil fuels utilised in electricity generation by fostering the use of enhanced, readily available, more efficient, less $CO_2$-emitting renewable fuel sources. The carbon produced by electricity production plants in the year 2030 would be approximately two gigatons, which is approximately one fifth less than in the Accelerated Transition Scenario (ATS). Cumulative investments in the ATS are similar to those in the Announced Pledges Scenario (APS), and investment funding requirements are not a constraint for China [23]. Some regions in China did not realise the gains from China's economic expansion and prosperity; however, the renewable sources supply chain, together with developing green technologies, will improve the entire country's economic growth [23]. The energy transition is welcomed and supported by labour since it has the potential to increase the employment rate. Employment growth may increase significantly if China is successful in securing part of the growing renewable energy market in countries that are not skilled in these emerging technologies. The advantage of increasing the range of the country's net-zero-$CO_2$ goal to include all GHG would result in the energy sector achieving net-zero carbon emissions much earlier than the year 2060 [23]. This will cater for other industries' $CO_2$ emissions that are not easily dispensed with. The government acknowledges that accelerated success in reducing carbon emissions through to the year 2030 is critical. Solar PV and wind energy of approximately 1.4 TW, which equates to approximately a fifth larger than the year 2050 APS, will drive compliance with the Paris Agreement [22].

In the Announced Pledges Scenario (APS), 40% of the country's electricity generation carbon emissions reductions by year 2060 are dependent on technologies that are currently being prototyped or in testing or demonstration phases and should be accessible as scheduled to replace present fossil fuel generation. It is envisioned that this would eradicate $CO_2$

emitted by the country's heavy industry (iron, steel, coal, etc.) by approximately 15% of the world's projected $CO_2$ financial allocation for a probability of 50% of limiting the average planetary increase in temperature to 1.5 °C [25].

China submitted their updated NDC in 2020 to the UNFCCC. The NDC includes the country's viewpoint and objectives for managing climate change and favourable outcomes obtained in the application of the NDC. Newfound actions to be executed in line with the revised NDC objectives include the country's aim to encourage global collaboration on climate change. China, similarly, is aiming to have its $CO_2$ emissions peak prior to the year 2030 and reach $CO_2$ neutrality prior to the year 2060 [5]. The NDC makes repeated reference to socio-economic transitions, unfortunately not supported by occupational health and safety, acceptable jobs or social conversation. The just transition of the JET pledge is not explicitly stated in the NDC; however, international cooperation on climate change was alluded to. The NDC states that "China is committed to promoting the establishment of a fair, reasonable, cooperative, and win-win global climate governance system and the global transition to green, low-carbon, climate-resilient and sustainable development" [7]. Training and education are covered appropriately in the NDC, although it does not adequately cover the requirement for skilled labour for a decarbonised economy. The strategy or plan to reskill current employees that are at risk because of their jobs becoming redundant from JET is not clear. China's need for upgrading professionals' labour force skills and green (renewable energy) skills is covered in education levels requirements [7].

## 4. Russia

### 4.1. Individual Indicators of Russia's Macroeconomics

The real GDP growth of Russia was 2.1% in 2022, and it is predicted to be 1.2% in the year 2023 [26]. Unemployment is at a record low of 3.5% and is forecasted to remain low to 2026 with the challenge of the Russia–Ukraine war. Russia's current-account surplus was declining rapidly to approximately 73% in the first quarter of 2023. However, increased oil and gas exports were at record highs in the year 2022. The Russian ministry reduced its forecast by 50% for the present unused budget in the year 2023 from $158 billion to $87 billion, with the commerce budget surplus prediction reduced from $228 billion to $152 billion [26].

Prior to the Russia–Ukraine war, Russia, like numerous resource-affluent and electricity-exporting countries, had begun JET; however, this progress slowed due to challenges to its implementation, such as the country's financial status, which is highly dependent on hydrocarbon export income. From 2000 to 2005, Russia managed to raise energy-resource exports substantially, i.e., by 56%, thereby accelerating economic growth and reinforcing Russia's position in the global arena as an energy superpower [27]. The worldwide economic recession in 2008 resulted in significantly reduced energy-resource exports. After the recession, the years 2011 to 2014 saw inflated oil costs as a result of poor international trade demand and declining petrodollar income, adversely affecting GDP (the oil price was $110 per barrel). Income from oil and gas global sales decreased from the high prices of 2008 to 2012, compounded by decreasing prices for hydrocarbons; however, in 2017, hydrocarbons supplemented 25% of GDP and represented 39% of Russia's planned income, 65% of export revenue and 25% of total funding to the Russian economy [28]. Global renewable energy objectives and the transition towards zero carbonization is causing a significant decrease in hydrocarbon export revenues in Russia and is exacerbated by the Russia–Ukraine war.

### 4.2. Russia's Energy Policy

Electricity is produced mainly from gas, which accounts for 47%; coal accounts for 16%; hydroelectric accounts for 17%; nuclear accounts for 18%; oil accounts for 1%; and renewables account for 1%, as illustrated in Figure 5. The country's tactical behaviour with respect to JET is paramount for Russia and the planet. Numerous features of macroeconomics, institutional frameworks in the power production industry and climate change

and technology plans or policies exist which define the Federation's strategy towards the energy transition [29].

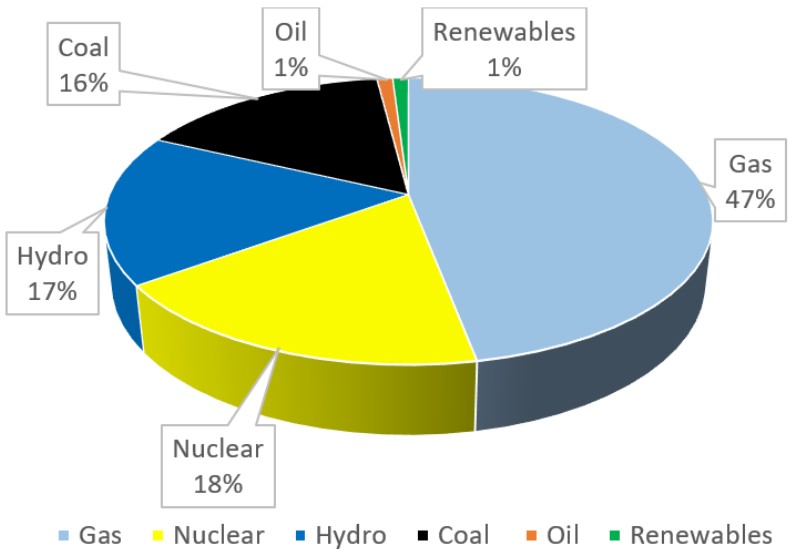

**Figure 5.** Current Russian energy mix.

Figure 6 illustrates the envisaged 2030 energy mix with the successful execution of the Energy Strategy of Russia (ESR, 2010): gas 36%, coal 5%, hydroelectric 8%, nuclear 4%, oil 39% and renewables 8% [29]. Russia ranked fourth in the world emissions rankings and is ranked as the fourth-largest primary energy consumer globally. Russia's climate-change-related policies are undergoing improvement, with opportunities to bridge gaps and develop the required strategies to improve climate–energy interface effectiveness [30]. Russia is committed to the United Nations Framework Convention on Climate Change (UNFCCC) and the Kyoto Protocol (Russia achieved its $CO_2$ aspiration of 5% below the 1990 level for the first commitment period between 2008 and 2012). Furthermore, the country supplies 10% of the world's energy fuel sources, 5% of international electricity usage and 16% of foreign commercial dealing in energy [30]. Russia is currently classified as the largest supplier of power fuels (ranked first in gas, second in oil and third in coal [29]. Taking cognisance of this fact, the country's planned intention with JET is mutually beneficial for national and international outcomes.

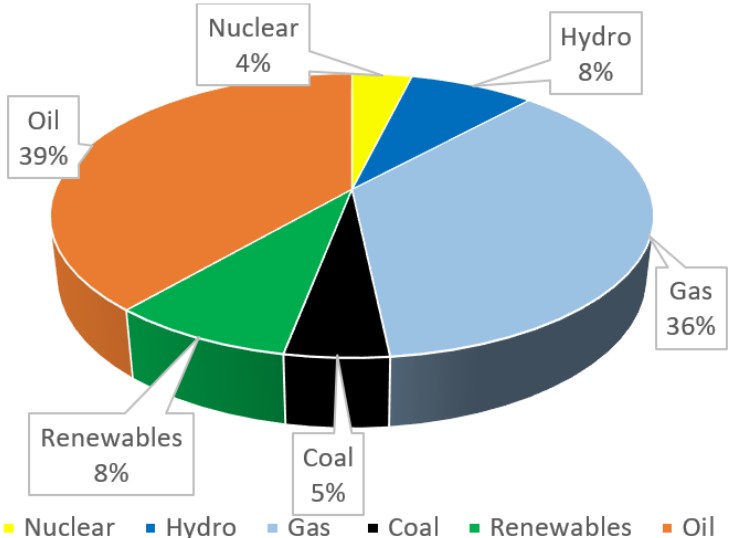

**Figure 6.** Russia's envisioned energy mix by 2030.

The 2022 Russia–Ukraine war has unfortunately negatively affected the global energy transition with financial and trade sanctions imposed on Russia by the European Union (EU) countries, the United Kingdom (UK) and the United States of America (USA). President Putin has halted the supply of coal, oil and gas to the countries that were dependent on these supplies in response to the abovementioned sanctions imposed on the country. Inevitably, this is causing an energy crisis, mostly for the EU countries, while the USA is using their oil reserves to dampen the effects of the crisis and prevent the increase in energy cost in their country. Other countries have had no choice but to refurbish and recommission their fossil fuel power plants to try to curb their energy challenges. This crisis has fortunately resulted in a positive spin-off, with short-term contingency plans to soften the impact on populations affected by the energy crisis, and governments are rapidly initiating longer-term actions: the United States of America's (USA) Inflation Reduction Act (IRA), the European Union's (EU) Fit for 55 package and REPowerEU, Japan's Green Transformation (GT) programme, Korea's intention to expand the portion of nuclear and other renewable sources in their energy mix and renewable aspirations in India and China [27].

The energy sector worldwide is under pressure to respond to the need for change away from hydrocarbons within the next 20 years. [27]. The Energy Research Institute of the Russian Academy of Sciences (ERI RAS) projects that with the unavoidable reshaping of intercontinental trade and declining requirements for the country's hydrocarbons, the foreign revenue from oil and gas to Russian GDP will be reduced by 18% between the years 2015 and 2040 [27]. The unfortunate consequences of climate-change-mitigating plans would adversely impact the country's financial stature. A decrease in economic growth results in growthless national energy needs, fixed sales costs, and a decline in the capital available for implementing the latest technology [27].

The Russian electricity sector has evolved into many private and state-owned companies [31]. State-owned companies make up the majority (approximately 70%), both in electricity production and in the power network. The Russian government owns the high-voltage power network of 220 kV and above, which Russia defines as the Transmission Network, and the majority of the electricity distribution networks. Approximately 60% of the country's electricity is produced from the combustion of fuel sources [31]. Russia pledged their commitment to the Paris Agreement in the year 2016, which included the development (by year 2020) of a plan for socio-economic expansion with a reduction in GHG by 2050 [30]. To soften the adverse impacts on the country's energy resources of compliance with the Paris Agreement, a co-ordinated strict legislative methodology is required to mitigate climate change effectively. From 1998 to 2008, GHG emissions increased by 16% over 11 years, which is considerably lower compared with GDP. Since the year 2017, electricity production has decreased its $CO_2$ impact ($g/CO_2$ per kWh) compared with countries such as Australia, China, India, Poland, Germany, Kazakhstan, the Arab countries of the Persian Gulf, the USA, Chile and South Africa [28,32]. This is because approximately 35% of electricity is generated from nuclear and hydropower plants with minimal $CO_2$ emissions [32]. Gas power plants contribute 48% of Russia's electricity capacity enabling the energy transition from petroleum and fossil fuelled electricity production (gas substitution into fossil-fuelled electricity production expanded by 5% from 2006 to 2017) [32]. The Paris Agreement and the Addis Ababa Agenda 2030 adversely impact the country's fossil fuel international trade (the natural gas and coal trade will be substantially affected by the zero-carbon emissions agreement globally by year 2050). Short-term exports are affected by the Russia–Ukraine war due to export and financial sanctions against Russia. Russia is confronted by impediments to international commerce for its energy-intensive products. Considering this point and low foreign revenue, national zero-$CO_2$ goals are being de-prioritised as it is challenging to attract investment with an imminent national recession and enforced embargos. This impacts energy cost negatively, with huge socio-economic implications for the Russian population, which substantiates the reasons for Russia's isolation from the global decarbonisation trend for many years [30].

Russia is proactive with regular climate change impact analyses for the Russian ecosystem and economy. Analysis shows that global warming may result in a few favourable effects on the country's financial status, with larger agricultural growth requirements and a declining need for heating, with some conceivable adverse effects that will arise during harsh weather conditions and permafrost liquefication, impacting the energy architecture [33]. Russia is focusing on aggressive and brave energy plans for improving energy efficiency as a result of increased interest in energy efficiency and energy saving from academic researchers and practitioners in Russia [34].Their studies have also shown that social obstacles impede the transformation towards an energy-efficient economy in the urban small business sector, as a result of disregard for recent energy-efficient technologies and of social hierarchical behaviour in large corporations that creates social barriers. To defeat these hurdles, organisations must introduce positive dynamics and internal educational plans to introduce energy management systems (EMS) [34]. Their increasing the contribution of renewable energy and revamping the energy industry shows that they are a responsible nation. The quantity of electricity needed to yield a unit of GDP (in buying capability equivalent terms) is currently more than 200% higher than the International Energy Agency (IEA) average, which re-emphasises the major inefficiencies that must be remedied. Russia has a mammoth capacity for energy efficiency, with a specialised potential of approximately 260 million tonnes of oil equivalent (Mtoe) [34].

Bringing this to fruition in the coming years will re-configure the country's energy equity, influencing upstream electricity production, thermal industry funding requirements, industrial competitive advantage and the availability of energy fuels for international commerce. Advancement in energy efficiency is conducive to relieving the effects of high energy prices for citizens. Energy efficiency can encourage a move towards cost-reflective pricing by inviting funding into the revamping of generation, transmission and distribution to reduce prices for end users [30].

Russia submitted an amendment in the year 2020 of its NDC to the UNFCCC, announcing their objective to restrict GHG emissions to 70% relative to 1990 amounts by 2030. The NDC considers the maximum possible permeable capacity of the forests and ecosystems and is contingent on the feasible and fair socio-economic growth of Russia [35]. With reference to the NDC, Russia's implementation initiatives concentrate on economic action to foster GHG decrease, raise energy efficiency in all areas and develop the use of non-fuel and renewable energy sources. Further mitigations focus on enhancing the condition of natural sinks and storing GHG [35]. It is reflected in the NDC that Russia will, furthermore, revise its GHG emission standards in synergy with global standards aimed at measuring the carbon footprint of producers. The NDC does not specifically mention a just transition in terms of the Just Energy Transition (JET); however, there are numerous references that summarise the quintessence of a just transition. It is noted that there is no specific reference to the value of jobs, employment creation, occupational health and safety (OHS), social shields or social discussion [35]. The NDC does not mention training, retraining or education of their labour force for the JET.

## 5. Brazil

### 5.1. Individual Indicators of Brazil's Macroeconomics

The real GDP growth of Brazil was 1.2% in the year 2022 and is predicted to be 1.6% in the year 2023, while consumption (use of goods and services) was 3.2% in the year 2022 [36]. The World Bank's conducting business ranking for Brazil in the year 2022 was 91 out of 190 countries. There was a significant increase in international tourism in Brazil by 107.5% in 2021 despite the COVID-19 pandemic, and this is predicted to grow by 61.5% in the years 2022 to 2023 [36].

### 5.2. Brazil's Energy Policy

Brazil has the greenest energy mix, with the use of renewable energy such as hydro, wind, solar, bioenergy and geothermal power plants which contribute 46% of the country's

energy requirements [37]. After China and Canada, Brazil is the third-largest hydroelectricity producer globally. Brazil's hydroelectric sector has almost unlimited potential, which has not been fully exploited given that the country has the capacity to construct numerous hydroelectric power plants [38]. A total electricity demand of 77% is produced from hydropower (hydro is 12% of the total energy source but is used as a major source to produce electricity), the rest being generated from coal, gas and other renewables in Brazil [38]. With its dependency on hydropower, Brazil is vulnerable to potential power supply shortages in drought years, as experienced during 2001 and 2002 when the country had severe droughts which resulted in an energy supply crisis [38].

The energy mix is depicted in Figure 7, where oil accounts for 36%, biomass accounts for 32%, hydro accounts for 12%, gas accounts for 11%, coal accounts for 5%, nuclear accounts for 2% and other renewables (wind, solar, etc.) account for 2% [37]. Brazil is the fifth-biggest country globally by ground area and is rated second amongst the top ten economies by magnitude of GDP [39]. Brazil is a major ally in commerce and at times a competitor of the USA with respect to energy fuels (e.g., petroleum and ethanol). The country was massively reliant on imports for its energy resources, but has now become an energy exporter that produces more energy than it consumes after 35 years of policy development to promote domestic energy resources (hydro, fossil fuels, biomass, gas, wind, nuclear and solar) [37]. Brazil is the tenth-biggest producer of electricity globally and the eighth-leading world user of energy [37]. Brazil's energy security has been significantly challenged throughout its history: major energy sources were inadequate including ethanol fuel for cars (at the beginning of the 1990s); droughts resulted in reduced hydropower (at the beginning of the 2000s); and expectations that the provincial energy from Bolivia and natural gas from Argentina would improve energy security were not met [39]. Bolivia's choice to nationalise their oil and gas sites surprised Brazil as a major setback, particularly as Brazil did not interpret the country as a dominion. The nationalization specifically shocked Petrobras, which was Brazil's oil company and had an approximately $1 billion financial stake in Bolivia [39]. Argentina added to the crisis by implementing policies that restricted the country's capacity to achieve its natural gas foreign commerce agreements. Being dependent on providers for a reliable energy source was identified as a risk to Brazil; however, by adopting better enabling strategies to encourage energy production and diversification, Brazil succeeded in achieving momentous growth through energy security advancements [39]. Argentina nationalised its energy resources as it recognised European and American organisations as wings of EU and U.S. dominion. Brazil, as a former colony, did not anticipate an associate Latin country to terminate Brazil's partnership, as Brazil did not deem the country as under dominion rule as with the likes of the United Kingdom, Spain and Portugal [39]. An incentivised reduced borrowing rate and funding plans, together with fair prices, was established for electric power bidding initiatives, which resulted in the wind power industry expanding substantially in the country.

Solar energy, as a source for distributed generation, is becoming an effective source of energy. The use of flexible fuel cars renders ethanol a widely used fuel. There was a blend mandate for 27.5% in gas including the opportunity of operating on 100% ethanol coupled with biodiesel mixing plans in the year 2018 [37]. Natural gas is a significant alternative and can be used as a transition fossil fuel until the year 2050, while other green energy sources are being phased in to decrease the use of fossil fuels. The Climate Transparency Policy Paper: Energy Transition in Brazil provides the generic energy plan for the country and determines initiatives, identifying gaps and prospects to accomplish greenhouse gas (GHG) emission alleviation objectives, providing policies and legislation to sustain vigorous evolution of the energy sector [37]. Brazil's renewable energy resources make up 46% of all primary energy resources and 85% of electricity generation. Currently, non-renewable energy sources comprise approximately 57% of the full primary energy supply, and 32.7% is utilised in the transport industry [37]. Brazilian energy intensity has been constant from the year 1990, below 4 TJ/million US$ [37]. Brazil's entire GHG production is approximately 1.6 billion tons of $CO_2$ and ranks 12th on the world emission rankings [40]. Power sector

investments were made in renewable energy from 2000 to 2013 due to structural reform in the energy sector. This led to changes in energy policy, which was more focused on the production of electricity by IPPs and thus increased usage of renewable energy from 48% to 51%. Wind power was the majority investment in the power sector between the years 2014 and 2016. Solar energy investments were augmented considerably to approximately 35% of overall funding in the power sector by 2016 [37].

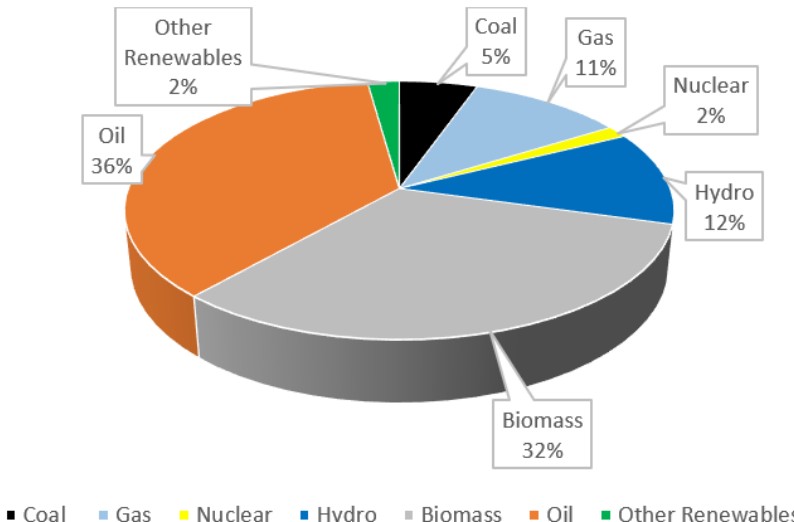

**Figure 7.** Brazil's current energy mix.

Funding authorised and approved for 2021 included fourteen solar photovoltaic plants (SPV), eight wind farms, two hydropower plants and one biomass-fired thermo-power plant, adding a further 880 MW capacity into the National Integrated System (SIN) [41]. Figure 8 shows the future envisaged energy mix improving to a greener energy mix with the additional capital injection of the Brazilian government executing their energy transition strategy. A total of $4.5 billion and 4040 employment opportunities are projected to be created, which is in compliance with ILO 2015. Wind power grew as a substitute aimed at the expansion of the electricity sector subsequent to the energy emergency of 2001 and is currently ninth in international volume (13 GW), with an average capacity factor of 40%, representing 8% of Brazil's electricity sector in terms of connected electrical power capacity.

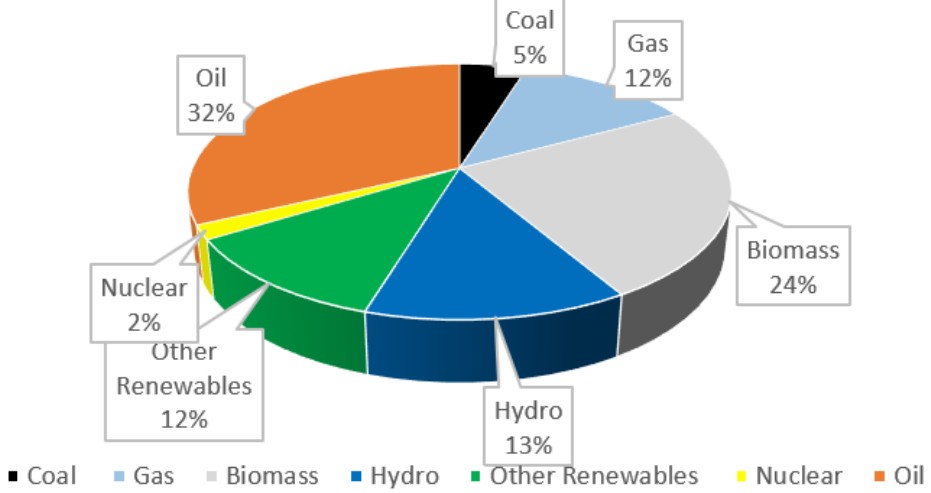

**Figure 8.** Brazil's envisioned energy mix by 2030.

The overall potential in Brazil is approximately 300 GW [41]. The Ministry of Mines and Energy has forecasted growth of 125% by 2026, when the predicted target of 28.6% wind power in the total capacity of the power pool in Brazil will be installed. It is estimated by the Brazilian Industrial Development Agency (ABDI) that by 2026 the wind power sector will produce approximately 200,000 novel permanent and ancillary employments [42,43]. The wind energy value chain has been rising due to the incentives offered by the government (tax exemptions, long-term financing, etc.). The country has considerably advanced their industrialised structure in the wind power sector to assemble the wind turbines, with the fabrication of numerous modules (towers, blades, subcomponents of the pivot and the nacelle) locally, hereby reducing the quantity of imported components compared to previous years [41].

The International Labour Organization (ILO, 2015) specified that if suitable strategies are implemented, the evolution of the global economy to a greener and more viable framework ought to create 620,000 additional employment opportunities in Brazil, which exceeds the 180,000 employment opportunities that may be lost (ILO, 2015). Funding of renewable energy resources, linked to incentive plans (tax reductions and attractiveness in financing) coupled with decreasing GHG discharge, may yield a large quantity of employment [44]. The wind power sector created 150,000 permanent jobs in the year 2016, at present totalling 13 GW, or 8.6% of the nationwide electricity sector [42]. The Light for All initiative is a notable achievement in pursuing the standardisation of available electrical power in pastoral areas, targeting old-fashioned inhabitants and regions of great scarcity. Started in 2003, it helped approximately 16 million people in 2016 [44]. An investment of $250 million in the year 2019 was intended to connect approximately 96,000 additional electrical connections in 17 states [45].

Numerous strategies were compiled in Brazil aimed at diversifying the energy mix through the Proalcool and Proinfa programs through its chief indicators [45]. These plans enabled the triumph of Brazil's biofuel and wind energy sectors, respectively. Public funding and incentives for substitute energy sources contributed an essential part to this progression and to launching the nation's wind energy value chain. Replicating these strategies is imperative for accelerating the progression of the solar energy value chain by integrating major funding in research and development. Brazil, with its Light for All program, has been fulfilling its social objectives [45]. In summary, renewable energies in the country are progressing swiftly and the energy transition will predictably evolve along with the nation's technical and financial progress [32].

Brazil revised its NDC in the year 2020 to reiterate the country's firm pledge to lower total GHG production by 37% by 2025 and by 43% by 2030 with reference to 2005 emissions. This would position Brazil to achieve climate neutrality (net-zero emissions) by 2060, as per the NDC. A further update was made in the form of a submission letter to comply with net-zero $CO_2$ emissions by the year 2050 [46]. This change was due to the sizable decrease in forestation, specifically in the Amazon Forest, between the years 2005 and 2015. The main contributor to $CO_2$ emissions is "slash and burn", related to land-clearing activities in the growth of agriculture [47]. The area specifically steering deforestation and farming happened to be the single area that experienced a GDP increase in the year 2020 [48]. By protecting the indigenous population's rights in particular, which are dependent primarily on the natural environment, the country is assisting and supporting them in adapting to the effects of environmental degradation affecting their livelihoods [49]. By securing the rights of these indigenous people, they may furthermore influence them to become ambassadors who promote JET as they frequently care for large quantities of natural reserves [50]. Another great benefit in securing land rights is that it provides the population that cares for the reserves with environmental service incentives as payment. This is a symbiotic relationship, permitting them to take part in preservation initiatives and be rewarded with money transfers in return [49]. Brazil has referenced the Brazilian constitution in the NDC, "paying due attention to the special needs of women and indigenous peoples" [51]. Another focus area is the skilling and reskilling of labour. Decreases in government funding for

training, the environment, science and technology are additional hindrances that stifle the growth of renewable energy skills [49]. Developing a greener economy cannot occur in isolation and will require major commitment and engagement from the public sector to promote the change to an achievable level, as comprehensive procedures will not occur freely through marketplace powers [49].

## 6. History of Energy Policy in SA

### 6.1. South Africa's Social and Economic Context in Comparison with the Rest of the BRICS Partners

South Africa makes up 1% of the world's land area with a population of 59 million people, which is 1.9% of the BRICS countries' total population, as shown in Table 1 [2]. The BRICS countries make up 30% of the world's territory. The average life expectancy of South Africans is 64 years, which coincidentally is the lowest amongst the BRICS countries [2]. South Africa's GDP growth rate is 2.1%, in comparison with an average GDP growth rate of 3.32% in the BRICS countries.

**Table 1.** BRICS countries' sizes, populations and GDP.

| Country | Country Area (1000 sq.km) | Population (Million Persons) | Life Expectancy (Years) | GDP Growth Rate % |
|---|---|---|---|---|
| Brazil | 8516 | 208 | 76 | 1.2% |
| Russia | 17,125 | 147 | 71.9 | 2.1% |
| India | 3287 | 1269 | 68.7 | 6.7% |
| China | 9600 | 1386 | 76.3 | 4.5% |
| South Africa | 1221 | 59 | 64 | 2.1% |

### 6.2. Individual Indicators of South Africa's Macroeconomics

Real GDP growth was 2.1% in the year 2022 and is predicted to be 4.8% in the year 2023, while consumption (use of goods and services) was 6.5% in 2022 [52]. The World Bank's business ranking for South Africain the year 2022 was 71 out of 190 countries. South Africa is rich in various natural resources (gold, diamonds, coal, iron ore, etc.) with the mining industry being one of the largest employers, providing the country with huge export revenue while supporting the local labour force. The large agricultural landscape of South Africa is also a crucial economic strength [52].

### 6.3. Current Energy Mix in South Africa

South Africa is the seventh-largest producer of coal, with most of the coal used to fuel fossil fuel power stations locally and internationally [53]. Coal, although used extensively as a fuel source, also contributes to many other industrial processes, such as steel production, cement processing, the paper and aluminium industries, chemical and pharmaceutical activities, coal gas and coal liquid for transportation energy and plant fertiliser, as detailed in Figure 9 [54]. South Africa traded 69 million tons of coal, valued at $5 billion, making it the country's fourth-most valuable mineral exported after gold ($201 billion), diamonds ($11 billion) and platinum ($8.5 billion), funding 4.5% of foreign income [53]. South Africa's domestic trade of approximately 183 million tons of coal created revenue more than 20% greater than for the coal exported [53]. Price, nonetheless, is not an acceptable justification for comparative tardiness. Cognisant of this fact, the coal mining labour force will be extremely concerned with the future reduction of coal usage in the generation of electrical power when it is replaced with cleaner and greener fuel sources, as it has a direct correlation with future job losses due to reduced coal-mining activity. If not managed sensitively and with appropriate engagement with labour, initiatives towards decarbonisation and most initiatives of the Just Energy Transition (JET) will be derailed [53].

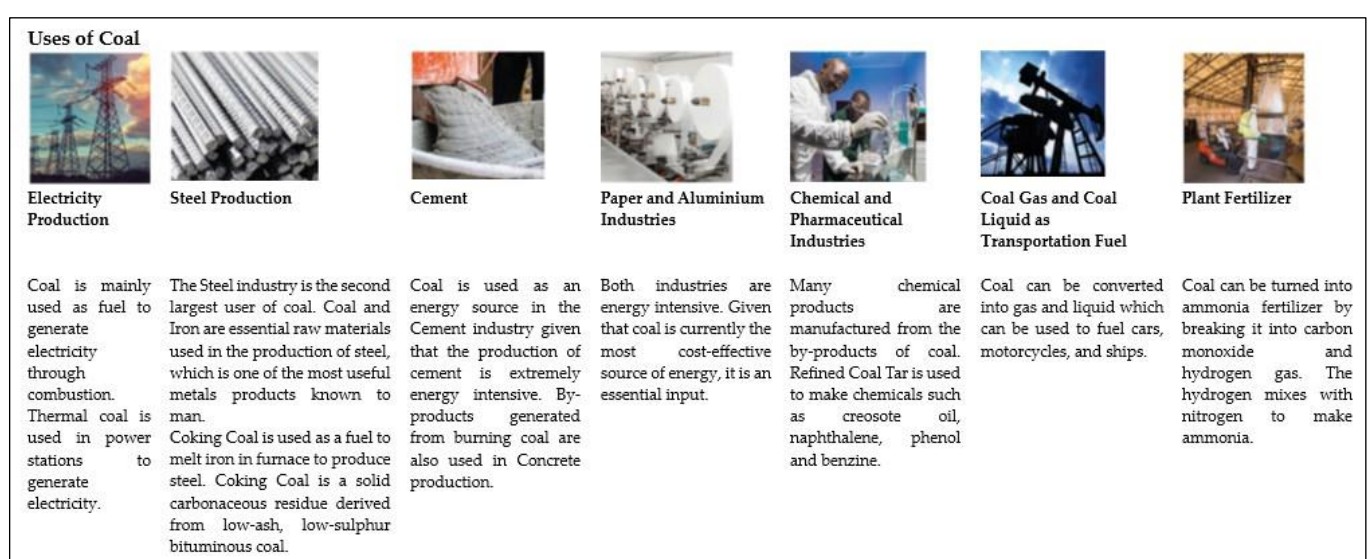

**Figure 9.** Uses of coal.

Coal is not an infinite resource, and demonstrating and reiterating this fact is crucial to JET and the drive to achieve the target of zero carbon emissions by 2050 to reduce global warming and climate change. Coal resources must be conserved to realistic levels, and the current global picture must be changed, as illustrated in Figure 10, where the present energy resources are 56% coal, 14% hydro, 10% wind, 8% solar, 7% gas, 3% biomass and 2% nuclear. It is apparent that coal is required for the future sustainability of various vital industries that are highly dependent on coal, including the pharmaceutical industry, which is overlooked due to the insignificant volumes of coal used in comparison with the large volumes consumed in the energy sector [55].

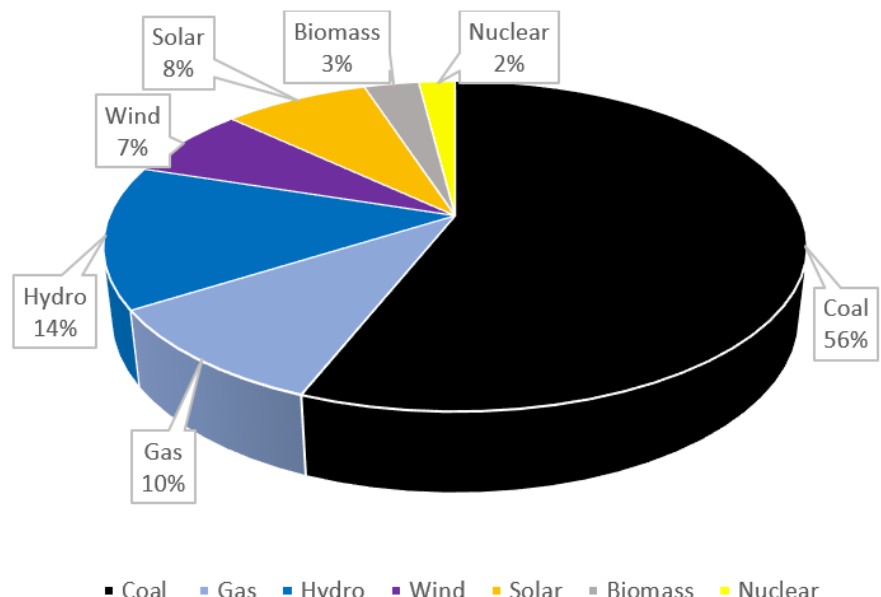

**Figure 10.** South African power plants' fuel sources.

In South Africa, most fossil fuel power stations are in their mid- to end-of-life phases with reference to the designed lifespan of 50 years, as illustrated in Figure 11 [56].

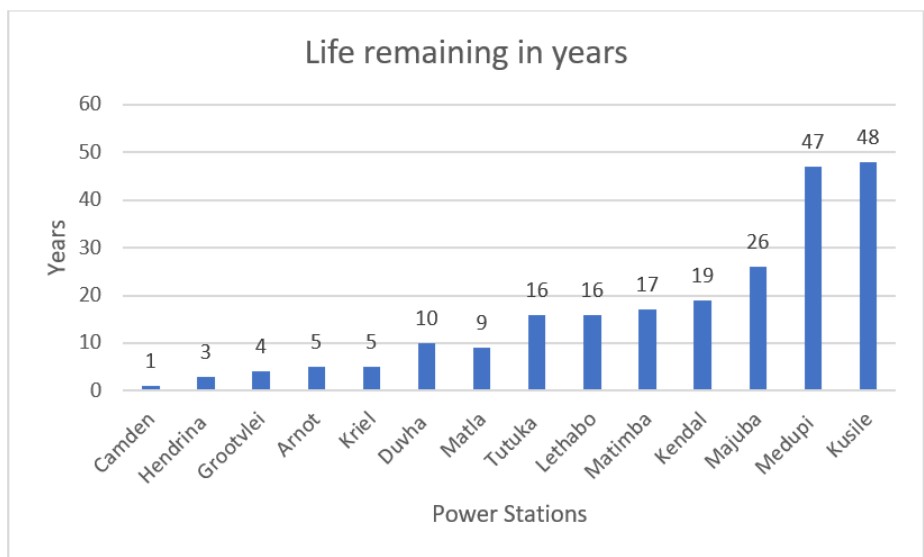

**Figure 11.** South Africa's fossil fuel power stations' economical life.

The decommissioning plan for coal-fuelled power plants, as shown in Table 2, coincides favourably with the country's Just Energy Transition to alternative lower-carbon fuel sources. Legislation and policies aligned to Just Energy Transition in South Africa must be formulated in a way that integrates related energy transition plans and strategies. Traditional or historical funding methodologies for the power utility business must change to accommodate the huge burden on the state and innovative investment models with public and private participation partnership models must be investigated. These models must be developed to allow local communities and citizens to contribute and share in future renewable and greener power plants. Participation or funding initiatives like trusts or equity-based crowdfunding can mitigate the effects of anticipated unemployment.

**Table 2.** South Africa's fossil fuel power plant decommissioning plans.

| Power Station | Decommissioning Date | Capacity (MW) |
|---|---|---|
| Arnot | 2021–2029 | 2232 |
| Camden | 2021–2024 | 1481 |
| Duvha | 2030–2034 | 2875 |
| Hendrina | 2021–2027 | 1638 |
| Kendal | 2038–2043 | 3840 |
| Kriel | 2026–2029 | 2850 |
| Lethabo | 2035–2040 | 3558 |
| Majuba | 2046–2050 | 3843 |
| Matimba | 2037–2041 | 3690 |
| Matla | 2029–2033 | 3450 |
| Tutuka | 2035–2040 | 3510 |
| **Total** | **2021–2050** | **32,967** |

*6.4. Insights into the Just Energy Transition in South Africa*

South Africa, as a developing country, initially did not commit to reducing carbon emissions. However, the Kyoto protocol and subsequent post-Kyoto protocol international initiatives aimed at the reduction of GHGs, to which South Africa has voluntarily committed itself, has had an influence on government with regards to energy policy [57,58]. This is in

the form of mandating the use of significantly more renewable energy sources, as seen in the Integrated Resource Plan (IRP 2010 and 2019) as well as, more specifically, addressing fossil fuels for electricity generation. The government took the stance in 2008 that all future fossil-fuel-fired power plants will only be licensed if they are carbon capture and sequestration (CCS)-compliant. This initiated a process of investigating viable renewable energy sources [57].

In South Africa, there are significant shortcomings with labour and urgent remedial action is required for compliance with the International Labour Organisation's (ILO) 2015 policies on environmentally friendly fuel sources and the adaptation of technologies to utilise single or multiple renewable fuel sources. There is also a huge necessity to bridge the gap in conventional funding methodology versus green funding imperatives [59]. Presently, the policies are not clear, but there are various collaborative initiatives with South African government departments (the Presidential Climate Commission (PCC), Labour, Environment, the Mineral Council of South Africa, Trade, and Industry and Competition), Eskom, Sasol, the Council for Scientific and Industrial Research (CSIR), the National Business Initiative (NBI), Business Unity South Africa (BUSA), and the private and public sectors, aimed at evaluating and charting a course and bridging gaps for the Just Energy Transition (JET). The Presidential Climate Commission (PCC) report provides confirmation that the South African government is pursuing the Just Energy Transition (JET) with conviction, but time will tell whether the transition will indeed be a just one. The Climate Investment Funds (CIF) announced the launch of the Accelerating Coal Transition (ACT) program in April 2021. ACT will be a catalyst in speeding up the transition from fossil fuels, predominantly coal, in a consistent and sequential approach, with governance, human resources and infrastructure, which are the three pillars of the program [60].

The CIF issued a call for Expressions of Interest (EOI), to which South Africa issued a response in June 2021, and it was favourably received in August 2021 with a proposal to concentrate on the decarbonisation of the electrical energy sector in the context of its Just Energy Transition (JET) initiatives. The CIF announced that South Africa's proposal was selected for the preparation of approximately \$500 million in funding by the government in cooperation with the World Bank (WB) and the African Development Bank (AfDB) in October 2021 [60]. The PCC completed South Africa's Just Energy Transition Investment Plan (JET-IP) in 2022. The JET framework focused on ensuring a just and equitable transition to net-zero $CO_2$ emissions in South Africa by 2050. The JET plans provide the required guidelines for various planning and policy-setting processes, ensuring the JET framework can achieve synergy across all JET planning in the country. The PCC commissioned numerous research studies with integrated public consultations to support the development of the JET plan in the year 2021. The studies and consultations aided in uncovering fundamental challenges to JET in South Africa, paving the way for a development of the JET framework that is practical, synchronised, executable and aligned to the requirements of all social partners [60]. The PCC report contains the findings of the assessments and consultations, emphasizing critical considerations for the development of the Just Transition framework. The information derived and extracted from each related study was used to develop a vision and to establish policies to support a just and equitable transition, supporting sustainable livelihoods, financing the transition, improving water security and creating good governance structures to manage JET [60].

These engagements are with reference to the 2015 ILO guidelines for a just transition towards environmentally sustainable economies and communities for all South Africans [6]. The ILO principles place emphasis on solid social confluence and pathways to energy sustainability [6]. Policies must conform, stimulate and realise necessary value and rights that are critical to energy sustainability. Clear policies across the economic, environmental, social, education/training and labour groups are essential to deliver an empowering environment for enterprises, labour, financiers and users to embrace and drive the evolution to environmentally sustainable and comprehensive economies and humanities [6].

South Africa's NDC was submitted to the UNFCCC in the year 2021, with a pledge for net-zero $CO_2$ emissions by 2050 [18]. In addition, the revised NDC caters for the country's economy, including "farming, forestry, other land uses, energy, industrial practices, product usage, waste, and five ($CO_2$, $CH_4$, $N_2O$, HFCs and PFCs)" types of gas management [61]. South Africa can be commended as it is the only BRICS member to categorically refer to and mention the Just Energy Transition in the country's NDC [61]. The NDC refers to the "just transition" as follows [61]:

- "In South Africa, a just transition is core to shifting our development pathway to increased sustainability, fostering climate resilient and low GHG emissions development, while providing a better life for all."

- "A just transition means leaving no-one behind. It requires procedural equity to lead to equitable outcomes. A just transition is at the core of implementing climate action in South Africa, as detailed in both the mitigation and adaptation goals presented below. As South Africa indicated at the UN Secretary General's Climate Action Summit in 2019, as part of ensuring a just transition we will need to put measures in place that plan for workforce reskilling and job absorption, social protection and livelihood creation, incentivising new green sectors of our economy, diversifying coal dependent regional economies, and developing labour and social plans as and when ageing coal-fired power plants and associated coal production infrastructure are decommissioned. Similar measures will be necessary to adapt to the impacts of climate change."

- "The just transition will also need international cooperation and requires solidary and concrete support. Ensuring that no one is left behind as we move from a high GHG emission, low employment energy development pathway to a low emission, climate-resilient and job-rich pathway, is central to our national work on development and climate change."

The Presidential Climate-Change Commission (PCC) was established for oversight over the country's JET. The NDC included the function and duty of the PCC. The PCC conducted major discussions over two years to develop the 2050 Vision and Pathways for a Just Transition and included in the JET decarbonisation and an economy and labour force that would be resilient to the transition [60]. South Africa will implement the 2050 Vision and Pathways for a Just Transition to minimise the temperature rise to 1.5 °C in compliance with the Paris Climate Change Agreement. At this stage, stakeholder engagements on JET have taken place predominantly at the national level, organised by the PCC and the National Planning Commission (NPC) [5]. The NPC has played a vital role with regard to Just Energy Transition (JET) policy development.

The NPC completed a multi-stakeholder social partner dialogue (SPD) for JET in the year 2019. The objective of the SPD was aimed at formulating a comprehensible national strategy that can be sufficiently funded and put into motion. The SPD outlined potential transition pathways in line with ILO's framework on the just transition [62]. These pathways relate to energy, land use and water. Mpumalanga and the Free State Province were prioritised as geographical locations for the first Just Transition implementation, being part of the Just Energy Transition (JET). In Mpumalanga, energy, water and land are interconnected, as the province has 80% of the fossil fuel power generation plants in the country. Hence, the JET immediately impacts Mpumalanga province and ensures that all three sectors are worked on simultaneously and in synergy. The Free State province will concentrate on water, land and, to a lesser extent, the gold mines [62].

Just Energy Transition Partnership (JETP)

The Just Energy Transition Partnership (JETP) is a voluntary funding partnership between developing and developed (or wealthier) countries that are highly dependent on fossil fuels, where developed countries support developing countries in accelerating decarbonisation and the transition towards clean energy with social justice in mind [63].

The South African (SA) government, Eskom (SA's power utility company) and a few developed countries (France, Germany, the United Kingdom (UK), the United States of

America (USA) and the European Union (EU)) had numerous engagements that resulted in the conceptualisation of the JETP. Initially, the concept concentrated on a Just Transition Transaction, as advocated by South African think tanks and Eskom. The requirements for financing mean that economic, technology and adverse social challenges converge for vastly fossil-fuel-reliant countries, as has become evident. The impetus was born of necessity, as Eskom, a deeply indebted power utility, needed to commission renewable energy power plants through a bespoke financial agreement that conventional climate finance did not clearly accommodate [63].

The first JETP, between South Africa and the International Partners Group (IPG), consisting of France, Germany, the United Kingdom, the United States of America, the European Union and the Climate Investment Funds (CIF), was announced at COP26 in Glasgow, UK, in 2021. This was inclusive of the JET execution plan for 2023 to 2027 in support of South Africa's decarbonisation plans [64]. In support of South Africa's JET initiative, the Independent Partners Group (IPG) will provide the country with a sum of $8.5 billion over a five-year period (2023 to 2027) to support its electrical power sector, initiate a green hydrogen sector and advance new energy vehicle (NEV) development [64].

The South African JETP is the most improved in terms of governance, negotiation, and investment configuration. It provides a framework for assessing the relationship between equity and Article 2.1(c) of the Paris Agreement, with its attention on justice and its proposal for bespoke finance arrangements, which can have a huge influence on aligning future finance flows to further the Paris Agreement's aims [63]. The JETP is led and directed by the purposefully formed Presidential Climate Finance Task Team directed by President Cyril Ramaphosa. The Presidency, South Africa, 2022, is the key interface between the South African government and the IPG. The financial proposal was considered, and the Just Energy Transition Investment Plan (JET IP) emerged via this process with the technical backing of a secretariat [64]. A political declaration by South Africa and the IPG made the initial proposal conditional to agreement on the investment framework, effectively rendering it a precondition for the provision of financing for South Africa's JETP [65].

## 7. Conclusions and Prospects for Further Research

A comparative study of the energy policies of the BRICS countries has shown that their populations' lifestyles, energy mixes, industries and commitments from their governments are extremely similar. South Africa is a small-scale reflection in many respects of Brazil, India and China as a developing nation with social and environmental responsibilities for a Just Energy Transition. Russia shares some similarities with its energy mix of fossil fuels being coal and natural gas. South Africa can replicate the hybrid or tailored energy policies of the BRICS countries in accordance with the South African landscape of energy sources readily available with the common objective of decarbonisation. The financial situation in South Africa is rather constrained compared with the BRICS countries but could be offset by the similar funding requirements to decommission and build new power plants. South Africa could apply and enter into negotiations for green finance and green bonds with the motivation of compliance with ILO 2015 and the Paris Agreement [6].

South Africa (SA), being part of BRICS, has the BRICS intergovernmental leverage to advance in the JET. SA was fortunate to be nominated as the chair of BRICS from January 2023. While the world is emerging from the adverse social and financial conditions of COVID-19, there is a war between Russia and Ukraine that has exacerbated the already-challenged global economy. This has resulted in a cascading effect on the bare necessities of energy and food shortages and has been exacerbated by trade and economic sanctions by the United Nations (UN), the United States of America (USA) and the European Union (EU) on Russia. Russia is a crucial leader in the BRICS alliance and is the third-largest oil producer, second-largest natural gas producer and top producer of steel and wheat globally. These consequences of the war and the sanctions against Russia have resulted in astronomical energy and food ecosystem demands due to this sudden constraint. As with any scarcity, there are higher prices as demand outweighs supply, and this has had an

unprecedented effect on food and energy. As a mitigation of this adverse impact, especially of the energy crisis in South Africa (SA) with regular load shedding, South Africa has developed the South African Economic Reconstruction and Recovery Plan (SAERRP). The objective of this plan is to aid the power or electricity sector to mitigate the challenges facing Eskom and the country to attain short- to long-term investment and eradicate the energy crisis, which is aligned to the country's NDC. SA, being a member and chair of the BRICS alliance, requested support in January 2023 through President Cyril Ramaphosa in its goal to alleviate the current and future energy crisis and the country's economic rehabilitation [66]. Coincidentally, the 2023 BRICS theme is Partnership for Mutually Accelerated Growth, Sustainable Development, and Inclusive Multilateralism [67]. BRICS member countries are fortunate in that they have vast natural resources and technology and are in a financial position to assist member countries.

The benefits of being complementary amongst its member countries was a pivotal point in the BRICS alliance at inception. BRICS has a framework in the form of a roadmap to achieve member countries' objectives. The BRICS Energy Research Cooperation Platform (ERCP) was initiated to attract and merge cooperation between subject matter specialists, organisations and research institutes to manage and expedite the similar ambitions of the BRICS in formulating and encouraging innovation in technology and governance [67]. So far, the progress of the first phase has been exceptional, with eight completed studies presented at the BRICS Ministers of Energy meetings, including a study on developing renewable energy and smart grids in the BRICS countries. Research on energy security is currently being executed by Russia, which is essentially the beginning of the second phase of the roadmap. The last phase is focused on building complementary beneficial cooperation, sharing best practices, using advanced BRICS technologies, trade and funding opportunities in individual BRICS member countries' economies [66].

There are vast economic and trade relations within the BRICS countries. Within the coal export market in South Africa, India acquires approximately 50% of the total coal exported annually by South Africa, and China has recently engaged in negotiations with SA for coal as well. The trade in oil and gas has been strengthened between Russia, China, India and Brazil in 2022, as has the renewable energy trade with South Africa [67]. There has been interest shown by China and India in the clean coal technologies of SA. ROSATOM, the Russian State Atomic Energy Corporation, agreed with SA to build a hydropower plant in the Mpumalanga province of South Africa. This would be the first step in a venture with Russia in SA to meet the JET priorities of net-zero $CO_2$ emissions, job creation, and stimulation of the economy and a start in resolving the country's electricity challenges with renewable energy [66]. The BRICS Energy and Green Economy Working Group's objective is to develop tangible deliverables for the JET. Currently, BRICS is engaging with the possibility of developing a BRICS African Centre of Excellence (ACOE) to further progress on the JET. The function of the ACOE would be to provide subject matter experts to research technological, social, economic and environmental components of the JET. It is acknowledged by the South African National Treasury that there are funds available to the tune of $12 trillion globally for green energy projects from various green-funding financial institutions. Developing countries, in particular, would be able to obtain this funding per Article 2.1(c) of the Paris Climate Change Agreement, although developing countries are seen as higher-risk. BRICS can also assist SA with obtaining these investments, despite policy impediments identified by South African regulatory bodies.

It is strongly recommended that we take a phased approach to form the basis of the next Integrated Resource Plan (IRP) for the country. The energy plan should assess the electrical power requirements of the country for the next 8 years, cognisant of end-of-life or near-end-of-life power plants up to 2030. The proposal for the next IRP should categorically state that South Africa will not invest in fossil fuels (coal, oil or natural gas) for new plants. Although natural gas is more efficient from a power plant perspective, there are still carbon emissions from these plants. CCS will make these gas plants less cost effective, as it will be a requirement to meet the target of net-zero carbon emissions by 2050. A drive to replace

power stations that are at the end of their economically and technically feasible lifespan with greener power alternatives, and the introduction of CCUS (carbon capture, usage and sequestration) to fossil fuel power stations that have 10 to 25 years productive lifespan, in South Africa would ensure a future with the energy mix changing in the right direction. It is necessary to rapidly increase the pace of the Just Energy Transition (JET) in South Africa to take the lowest-cost options to benefit the country both environmentally and economically while ensuring sustainable employment opportunities with readily available technologies.

The focus would be to grant approval to the state-owned utility company, Eskom, the private sector or both to replace the approximate capacity of 8201 MW of the retiring power plants by 2030 with alternative renewable energy plants in the most effective and efficient manner [68]. The fastest and most cost-effective technology currently would be solar PV coupled with battery energy storage systems (BESS) to replace the majority of the capacity requirement (proven in China and cognisant that China produces most of the solar panels and auxiliary equipment). BESS/solar PV can be hybridised to incorporate aquaponics or agrivoltaics, and the latter can be assessed for climate considerations and economics. For the newer electricity-generating power plants (with a lifespan of 20 years or more), the feasibility of retrofitting for carbon capture, usage and sequestration (CCUS) or at least carbon capture and sequestration (CCS) must be economically evaluated to assess their viability [69]. This would provide a support and buffer while the SA energy sector researches and assesses the viability of hydrogen, nuclear, wind and gravitational power generation based on their effectiveness and ultimately their efficiency for reducing global carbon emissions and producing electrical power.

*Prospects for Future Research*

Future research must be conducted into the viability and feasibility of a consolidated BRICS JET policy with embedded shared renewable energy resources and components to encourage the BRICS member countries to transition to zero carbon, in compliance with the Paris Climate Change Agreement. The shared resources would include agricultural expansion with cross-subsidization and funding within BRICS to encourage the production of vast amounts of food to sustain the nutritional requirements of the human population and not be limited to the BRICS member countries. This may lead to the BRICS sustainability alliance being far greater than just JET.

**Author Contributions:** R.R., research study, assessment and compilation of the paper. N.M., direct review and comment as academic supervisor. L.N., advice, review and comment as academic supervisor. All authors have read and agreed to the published version of the manuscript.

**Funding:** This research received no external funding. The APC was funded by Prof Lwazi Ngubevana, University of Witwatersrand.

**Institutional Review Board Statement:** Not applicable.

**Informed Consent Statement:** Not applicable.

**Conflicts of Interest:** The authors declare no conflict of interest.

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
