# Peer review of "A Review of the Energy Policies of the BRICS Countries: The Possibility of Adopting a Just Energy Transition for South Africa"

_sustainability, doi:10.3390/su16020703_

Round 1

Reviewer 1 Report

Comments and Suggestions for Authors

This manuscript reviewed the BRICKS energy status and policy, and recommend  the development direction for South Africa. In general, it is well organized and results have value for readers. It can be accepted after revision.

1: line 39-44, JET rankings and ETI ratings of the BRICS countries, the number meanings, the higher, the better or reverse?

2: line 69, The Prime Minister of Indian government, Narendra Modi, announced  

3: line 83, please confirm this value, 516MW, or 516GW?

4: line 91, does air quality have direct relation with electricity  is affordable  or not? what is affordable electricity? fossil fuel or biomass? unavailability means bad efficiency or emission control?

5: line 91, what features of distribution utility companies resulted the poor quality? low efficiency or low level technology

6: GDP, such abbreviation, the full expression only is needed at the first appearance.

Comments on the Quality of English Language

The expression is bit hard to read and understood, many nouns used as adjective, hard to find subject.

Author Response

1: line 39-44, JET rankings and ETI ratings of the BRICS countries, the number meanings, the higher, the better or reverse?

Adressed , highlighted in yellow in text

2: line 69, The Prime Minister of Indian government, Narendra Modi, announced 

Corrected, Line 69 is now line 111 , (highlighted in yellow in text)

3: line 83, please confirm this value, 516MW, or 516GW?

Its corrected to 516GW, line 83 is now line 123 (highlighted in yellow in text)

4: line 91, does air quality have direct relation with electricity  is affordable  or not? what is affordable electricity? fossil fuel or biomass? unavailability means bad efficiency or emission control?

Addressed in line 132 (highlighted in yellow in text)

5: line 91, what features of distribution utility companies resulted the poor quality? low efficiency or low level technology

Explained in line 138 (highlighted in yellow in text)

6: GDP, such abbreviation, the full expression only is needed at the first appearance.

Noted and revised accordingly

Reviewer 2 Report

Comments and Suggestions for Authors

The topic of the manuscript is relevant, and its content may be of interest to potential readers. However, there are some of gaps and shortcomings that authors need to remove before a manuscript can be published:

1. I have some doubts about the correctness of the type definition of this manuscript (Review). Detailed requirements for writing a Review are presented on the MDPI website (https://www.mdpi.com/about/article_types):

Reviews offer a comprehensive analysis of the existing literature within a field of study, identifying current gaps or problems. They should be critical and constructive and provide recommendations for future research. No new, unpublished data should be presented. The structure can include an Abstract, Keywords, Introduction, Relevant Sections, Discussion, Conclusions, and Future Directions, with a suggested minimum word count of 4000 words”.

Are the authors sure that the content of their manuscript fully meets these requirements? After all, the authors do not so much analyze the literature as describe energy policy and energy consumption in different countries.

2. In connection with the previous remark, I suggest that the authors analyze a larger number of scientific articles on the topic of the manuscript, but carry out such an analysis separately for each of the four countries (India, China, the Russian Federation, and Brazil). Then, it is necessary to analyze the literature on South Africa and, ultimately, to establish what experiences of the other four countries South Africa can use. In part, the authors have done all this, but it is not enough for the manuscript to meet the requirements for the Review. Therefore, it is necessary to analyze a larger number of literary sources.

3. Ultimately, more attention needs to be paid to the gaps in the literature and further research directions on the topic of this manuscript (in particular, research on energy consumption, energy transition and energy policy in South Africa). Then the main requirements for the Review Article will be fulfilled.

4. In the penultimate paragraph of the introduction, it is worth stating the purpose and objectives of this scientific work.

5. In the last paragraph of the introduction, it is advisable to present the structure of the manuscript more clearly. It is advisable to start this paragraph as follows: "Further material is divided into several parts. Thus, in Section 2……. Section 3 presents……".

6. I ask the authors to check the correctness of the names of all sections, subsections, figures and tables. For example, section 2 and subsection 2.2 have the same title. Figure 10 is missing the name of the country. Please note that all figures and tables must be referenced in the text. At the same time, figures and tables should be placed after references to them.

7. Authors should check the compliance of the manuscript with the design requirements. Special attention should be paid to the design of the tables and the list of sources.

8. At the end of the manuscript, it is necessary to provide data on the contributions of the authors and other information. I recommend taking already published sexes in this magazine as samples.

9. The authors consider the energy policies of the BRICS countries. Can the authors describe in the Conclusions the role that this intergovernmental organization plays or can play in ensuring the energy transition in the countries that are its members? Will the announced expansion of BRICS affect this?

10. Grammar and style should be improved. Some sentences could be worded better. For example, at the beginning of the Introduction, the history of the creation of BRICS should be described more clearly. Since what year has this organization been made up of five countries (line 28)?

I think it is appropriate to acquaint the authors with these comments, suggestions and questions. I hope that such an acquaintance helps to improve the quality of the manuscript, which is expected to be published in such a high-ranking journal as "Sustainability".

Comments on the Quality of English Language

Grammar and style should be improved. Some sentences could be worded better.

Author Response

  1. I have some doubts about the correctness of the type definition of this manuscript (Review). Detailed requirements for writing a Review are presented on the MDPI website (https://www.mdpi.com/about/article_types):

“Reviews offer a comprehensive analysis of the existing literature within a field of study, identifying current gaps or problems. They should be critical and constructive and provide recommendations for future research. No new, unpublished data should be presented. The structure can include an Abstract, Keywords, Introduction, Relevant Sections, Discussion, Conclusions, and Future Directions, with a suggested minimum word count of 4000 words”.

Are the authors sure that the content of their manuscript fully meets these requirements? After all, the authors do not so much analyze the literature as describe energy policy and energy consumption in different countries.

Authors reviewed the definition and agree to the Review.

  1. In connection with the previous remark, I suggest that the authors analyze a larger number of scientific articles on the topic of the manuscript, but carry out such an analysis separately for each of the four countries (India, China, the Russian Federation, and Brazil). Then, it is necessary to analyze the literature on South Africa and, ultimately, to establish what experiences of the other four countries South Africa can use. In part, the authors have done all this, but it is not enough for the manuscript to meet the requirements for the Review. Therefore, it is necessary to analyze a larger number of literary sources. Noted recommendations, studied more literature and revised accordingly in journal (highlighted in yellow).
  2. Ultimately, more attention needs to be paid to the gaps in the literature and further research directions on the topic of this manuscript (in particular, research on energy consumption, energy transition and energy policy in South Africa). Then the main requirements for the Review Article will be fulfilled. Was noted and revised in the journal (highlighted in yellow of text)
  3. In the penultimate paragraph of the introduction, it is worth stating the purpose and objectives of this scientific work. Was added in lines 79 - 84 as advised.
  4. In the last paragraph of the introduction, it is advisable to present the structure of the manuscript more clearly. It is advisable to start this paragraph as follows: "Further material is divided into several parts. Thus, in Section 2……. Section 3 presents……".Added as advised in lines 86-92
  5. I ask the authors to check the correctness of the names of all sections, subsections, figures and tables. For example, section 2 and subsection 2.2 have the same title. Figure 10 is missing the name of the country. Please note that all figures and tables must be referenced in the text. At the same time, figures and tables should be placed after references to them. Corrected
  6. Authors should check the compliance of the manuscript with the design requirements. Special attention should be paid to the design of the tables and the list of sources. Checked accordingly.
  7. At the end of the manuscript, it is necessary to provide data on the contributions of the authors and other information. I recommend taking already published sexes in this magazine as samples. Added
  8. The authors consider the energy policies of the BRICS countries. Can the authors describe in the Conclusions the role that this intergovernmental organization plays or can play in ensuring the energy transition in the countries that are its members? Will the announced expansion of BRICS affect this? Was included in the conclusion
  9. Grammar and style should be improved. Some sentences could be worded better. For example, at the beginning of the Introduction, the history of the creation of BRICS should be described more clearly. Since what year has this organization been made up of five countries (line 28)?Was included in line 28 and overall document checked for grammar.

I think it is appropriate to acquaint the authors with these comments, suggestions and questions. I hope that such an acquaintance helps to improve the quality of the manuscript, which is expected to be published in such a high-ranking journal as "Sustainability".

Reviewer 3 Report

Comments and Suggestions for Authors

  This review summarizes the energy policies of the BRICS countries in detail and presents a possible just energy transition policy adoption for South Africa. It is significant and well-organized and can be accepted in its current form. 

• What is the main question addressed by the research? It proposes possible energy policies for South Africa by learning from the Just Energy Transition (JET) policies of the BRICS countries.   • What specific improvements should the authors consider regarding the methodology? What further controls should be considered? To summarize the history of energy consumption. transition, and policies in South Africa; The government control to implement the energy policies.   • Are the conclusions consistent with the evidence and arguments presented and do they address the main question posed? Yes, it raises a possible pathway to reduce carbon emissions i.e. solar PV.
• Are the references appropriate? Yes, but adding more references (5-10) will make the review more comprehensible.  

• Please include any additional comments on the tables and figures.

Figure 11 has a negative value in service life. It needs elaboration or fixing of the y-axis.

Author Response

This review summarizes the energy policies of the BRICS countries in detail and presents a possible just energy transition policy adoption for South Africa. It is significant and well-organized and can be accepted in its current form. 

• What is the main question addressed by the research? It proposes possible energy policies for South Africa by learning from the Just Energy Transition (JET) policies of the BRICS countries.  

• What specific improvements should the authors consider regarding the methodology? What further controls should be considered? To summarize the history of energy consumption. transition, and policies in South Africa; The government control to implement the energy policies.  Was done in text

• Are the conclusions consistent with the evidence and arguments presented and do they address the main question posed? Yes, it raises a possible pathway to reduce carbon emissions i.e. solar PV.
• Are the references appropriate? Yes, but adding more references (5-10) will make the review more comprehensible.  Nineteen more references was added.

• Please include any additional comments on the tables and figures.

Figure 11 has a negative value in service life. It needs elaboration or fixing of the y-axis.  Noted , and corrected 

Reviewer 4 Report

Comments and Suggestions for Authors

Dear Authors,

- The item "Introduction" is not well developed and does not problematize the theme to give theoretical support for the proposed objective;

- There is no minimum description of methodological procedures;

- The results do not explore sufficiently to the regional characteristics of each of the countries. In my view, It is just a review without any density about the power matrix and some projections to obtain the JET;

- Considering these aspects, my decision is to reject this manuscript.

Comments on the Quality of English Language

Minor editing of English language required.

Author Response

The item "Introduction" is not well developed and does not problematize the theme to give theoretical support for the proposed objective; Addressed in lines 79-84

- There is no minimum description of methodological procedures; Shown in lines 86-92

- The results do not explore sufficiently to the regional characteristics of each of the countries. In my view, It is just a review without any density about the power matrix and some projections to obtain the JET; Highlighted in yellow in the text and especially captured in the conclusion 

Reviewer 5 Report

Comments and Suggestions for Authors

The authors have crafted a superb review. The only pending matter prior to its publication is the need to improve the resolution of all figures.

For a review paper, it is highly recommended to delve into a detailed discussion and comparison of the results in the specified section.

Author Response

The authors have crafted a superb review. The only pending matter prior to its publication is the need to improve the resolution of all figures. Revised accordingly

For a review paper, it is highly recommended to delve into a detailed discussion and comparison of the results in the specified section. Provided in conclusion , highlighted in yellow.

Thanks for your kind comments .Much appreciated.

Round 2

Reviewer 2 Report

Comments and Suggestions for Authors

In my opinion, the text of the manuscript has improved. The authors took into account a number of my comments. At the same time, the text of the manuscript still has certain shortcomings (mostly of a technical nature) and some debatable points, namely:

1. In my opinion, the style and grammar can still be improved. Some sentences could be worded better. For example, this applies to the second sentence of the Abstract (lines 13-15). It is also worth improving the description of the structure of the manuscript (lines 87-92). You can write something like this: “Further material is divided into several parts. The next four sections provide an overview of the energy policies of BRICS member countries such as India (Section 2), China (Section 3), the Russian Federation (Section 4) and Brazil (Section 5). Section 6 presents the findings of a review of energy policy in South Africa in the context of ensuring a just energy transition. Finally, in Section 7, conclusions are given and prospects for further research are described."

In lines 219, 335, 485, you should indicate the names of the relevant countries (GDP of which country?).

Titles of sections and subsections could be formulated better and more clearly. For example, section 2.1 “India's Macroeconomics” might be better titled “Individual Indicators of India's Macroeconomics” since there are quite a few macroeconomic indicators in general.

Therefore, I believe that authors should do a final grammar check of the entire text of the manuscript, starting with the abstract and ending with the conclusions.

2. Given that this article has a review character (type of article - Review), in the last section, more attention should be paid to the prospects of further scientific research on this topic. I already noted this point in my previous review, but the authors did not take my point into account enough. In particular, the authors paid insufficient attention to identifying gaps in the literature. It would be worthwhile to devote at least one (last) paragraph of section 7 to a description of the prospects for further research. Then perhaps section 7 should be called "Conclusions and prospects for further research".

3. The design of the manuscript needs improvement. In particular, this applies to paragraph indentation, spacing between paragraphs, design of tables and list of sources, etc. All tables and figures should be placed after references to them in the text of the manuscript. I can't find the reference for Figure 6 and Figure 10. Perhaps the authors should take as a sample those articles that have already been published in this journal.

Comments on the Quality of English Language

In my opinion, the style and grammar can still be improved. Some sentences could be worded better.

Author Response

In my opinion, the text of the manuscript has improved. The authors took into account a number of my comments. At the same time, the text of the manuscript still has certain shortcomings (mostly of a technical nature) and some debatable points, namely:

  1. In my opinion, the style and grammar can still be improved. Some sentences could be worded better. For example, this applies to the second sentence of the Abstract (lines 13-15). (Corrected accordingly.) It is also worth improving the description of the structure of the manuscript (lines 87-92). You can write something like this: “Further material is divided into several parts. The next four sections provide an overview of the energy policies of BRICS member countries such as India (Section 2), China (Section 3), the Russian Federation (Section 4) and Brazil (Section 5). Section 6 presents the findings of a review of energy policy in South Africa in the context of ensuring a just energy transition. Finally, in Section 7, conclusions are given and prospects for further research are described." Revised as advised, Thanks

In lines 219, 335, 485, you should indicate the names of the relevant countries (GDP of which country?). Done as advised.

Titles of sections and subsections could be formulated better and more clearly. For example, section 2.1 “India's Macroeconomics” might be better titled “Individual Indicators of India's Macroeconomics” since there are quite a few macroeconomic indicators in general. Corrected.

Therefore, I believe that authors should do a final grammar check of the entire text of the manuscript, starting with the abstract and ending with the conclusions. Completed.

  1. Given that this article has a review character (type of article - Review), in the last section, more attention should be paid to the prospects of further scientific research on this topic. I already noted this point in my previous review, but the authors did not take my point into account enough. In particular, the authors paid insufficient attention to identifying gaps in the literature. It would be worthwhile to devote at least one (last) paragraph of section 7 to a description of the prospects for further research. Then perhaps section 7 should be called "Conclusions and prospects for further research". Noted and included as advised.
  2. The design of the manuscript needs improvement. In particular, this applies to paragraph indentation, spacing between paragraphs, design of tables and list of sources, etc. All tables and figures should be placed after references to them in the text of the manuscript. I can't find the reference for Figure 6 and Figure 10.Perhaps the authors should take as a sample those articles that have already been published in this journal. Was corrected accordingly, Thanks 

The authors would like to thank you for your patience and most valuable comments to improve the manuscript. It much appreciated.

Reviewer 4 Report

Comments and Suggestions for Authors

Dear Authors,

I checked that you made all suggestions and improve your manuscript. My decision is "accept in present form". 

King regards.

Author Response

Dear Authors,

I checked that you made all suggestions and improve your manuscript. My decision is "accept in present form". 

King regards.

The authors would like to thank you for your valuable time taken to review and comment and accept the manuscript. Much appreciated.

Reviewer 5 Report

Comments and Suggestions for Authors

The authors carefully revised the MS. It can be accepted.

Author Response

The authors carefully revised the MS. It can be accepted.

The authors would like to thank you for your valuable time taken to review, comments and accept our manuscript. Much appreciated.